# Network approaches and interventions in healthcare settings: A systematic scoping review

**Ameneh Ghazal Saatchi**[1]*, **Francesca Pallotti**[2], **Paul Sullivan**[3,4]

**1** Imperial College London, London, United Kingdom, **2** Department of Business, Operations and Strategy, University of Greenwich, London, United Kingdom, **3** NIHR ARC Northwest London, Imperial College London, London, United Kingdom, **4** University Sussex Hospitals NHS Foundation Trust, Sussex, United Kingdom

* AGSaatchi@gmail.com

## Abstract

### Introduction

The growing interest in networks of interactions is sustained by the conviction that they can be leveraged to improve the quality and efficiency of healthcare delivery systems. Evidence in support of this conviction, however, is mostly based on descriptive studies. Systematic evaluation of the outcomes of network interventions in healthcare settings is still wanting. Despite the proliferation of studies based on Social Network Analysis (SNA) tools and techniques, we still know little about how intervention programs aimed at altering existing patterns of social interaction among healthcare providers affect the quality of service delivery. We update and extend prior reviews by providing a comprehensive assessment of available evidence.

### Methods and findings

We searched eight databases to identify papers using SNA in healthcare settings published between 1st January 2010 and 1st May 2022. We followed Chambers et al.'s (2012) approach, using a Preferred Reporting Items for Systematic reviews and Meta-Analyses extension for Scoping Reviews (PRISMA-ScR) checklist. We distinguished between studies relying on SNA as part of an intervention program, and studies using SNA for descriptive purposes only. We further distinguished studies recommending a possible SNA-based intervention. We restricted our focus on SNA performed on networks among healthcare professionals (e.g., doctors, nurses, etc.) in any healthcare setting (e.g., hospitals, primary care, etc.). Our final review included 102 papers. The majority of the papers used SNA for descriptive purposes only. Only four studies adopted SNA as an intervention tool, and measured outcome variables.

### Conclusions

We found little evidence for SNA-based intervention programs in healthcare settings. We discuss the reasons and challenges, and identify the main component elements of a

**Data Availability Statement:** All relevant data are within the manuscript and its Supporting Information files.

**Funding:** The authors received no specific funding for this work.

**Competing interests:** The authors have declared that no competing interests exist.

network intervention plan. Future research should seek to evaluate the long-term role of SNA in changing practices, policies and behaviors, and provide evidence of how these changes affect patients and the quality of service delivery.

## Introduction

It is widely recognized that there is a gap between best achievable healthcare outcomes and those that are actually delivered, even in the best funded systems, suggesting that more is required than simply increasing available resources [1, 2]. Improving healthcare outcomes requires changes in frontline clinical practice, which in turn involves the ability to disseminate information across diverse teams, and to engender alignment of multiple groups.

The diffusion of practices and behaviors within any healthcare setting may be usefully framed as a network problem involving multiple individuals and the way they relate and interact with one another. Leaders aiming to improve healthcare outcomes would benefit from understanding how team members interact, and how interactions may be leveraged to optimize the adoption and diffusion of new practices. Information about patterns of interaction can be obtained using Social Network Analysis (SNA). SNA provides a set of tools and techniques used to investigate structural characteristics of networks [3], and understand how a broad range of behaviors may be triggered by social interaction [4]. SNA generates three main types of outputs. The first is a visual representation of networks structures, or network graphs. The second is a set of metrics providing quantitative information on properties of networks, such as density, or properties of individuals, such as centrality. The third type of output is produced by statistical models for network data, such as models for the analysis of longitudinal networks [5].

SNA outputs can be used to inform the design, implementation and monitoring of behavioral change programs, policies and practices [4, 6]. A network intervention can be defined as a structured process using social networks to accelerate behavior change or improve organizational performance [7]. Social networks are channels for information diffusion and interpersonal influence. Hence, changing the wiring of an existing social network may determine changes in how behaviors, ideas and practices spread in a social group.

Valente [7] proposed a taxonomy of four types of network intervention strategies: i) 'Individuals', based upon the identification of individuals with certain network characteristics who are recruited to act as change proponents; ii) 'Segmentation', involving the identification of subgroups in a network on which to focus behavioral change; iii) 'Alteration', whereby an existing network is changed by adding or removing ties or nodes in order to alter patterns of interaction and diffusion, and finally iv) 'Induction', whereby peer-to-peer interactions are encouraged through, for example, the use of meetings or training events bringing previously unconnected people together.

While a large body of research is available that relies on SNA to examine networks of health professionals in healthcare settings, much of this research has been descriptive, with limited reporting of the relationship between network interventions and clinical or organizational outcomes. This is confirmed by recent systematic reviews. For example, Chambers et al.'s [8] systematic scoping review of SNA-based studies in healthcare settings found very little evidence of the use of SNA as part of an intervention. Cunningham et al.'s [9] review (1995–2009) included 40 eligible studies. Only one described an SNA-based intervention using survey data to identify opinion leaders, but did not measure its impact. Bae et al.'s [10] systematic review

included 28 eligible studies (up to 2013), none of which reported on outcomes of SNA-based interventions. A recent umbrella review by Hu et al. [11] included 13 reviews between 2010 and 2019 and demonstrated a wide applicability of SNA to study health professional networks. Of the 330 papers included in the reviews, only one reported on a network intervention.

The aim of the present review is threefold. First, provide an update of prior reviews by searching for papers using SNA to investigate networks of healthcare professionals in healthcare settings. Second, identify research reporting about network-based interventions and their outcomes. Third, identify the component elements and discuss the main challenges of a network intervention strategy to call attention on its potential in healthcare settings. The primary research question that this review seeks to address is what evidence is available on the adoption of network interventions and evaluation of their effect on care processes and outcomes.

## Methods

### Protocol

The literature review was undertaken in accordance with the protocol (S1 File) followed by Chambers' et al. in their 2012 review [8]. We used the Preferred Reporting Items for Systematic reviews and Meta-Analyses extension for Scoping Reviews (PRISMA-ScR) statement and guidelines (S2 File) [12].

### Information sources and search strategy

The literature search focused on identifying studies performing SNA on networks of healthcare professionals in healthcare settings. We used the same search strategy, inclusion and exclusion criteria and keywords as those used by Chambers et al. [8]. We performed a systematic electronic database search of OVID MEDLINE (R) ALL first, using free text terms, synonyms and subject headings associated with social networks and the methods used to investigate them including 'sociometrics', 'sociograms' and 'sociomaps'. We also used words associated with SNA software, such as NetDraw and UCINET. Finally, the search strategy included the subject headings inter-professional relations, inter-disciplinary communication and physician-nurse relationships. The search strategy was later adapted for other databases in our search. Specifically, for the period 1st January 2010 to 1st May 2022, we searched the following databases: OVID MEDLINE (R) ALL, EMBASE Classic+EMBASE, APA PsycINFO, Health Management Information Consortium (HMIC), the Cochrane Library (Cochrane Database of Systematic Reviews, Cochrane Protocols and Cochrane Central Register of Controlled Trials), CINAHL Plus, Business Source Ultimate, Social Science Citation Index (SSCI) and Conference Proceedings Citation Index—Social Science & Humanities (CPCI-SSH) databases. Reference lists of relevant reviews and studies were searched, as was the website of the International Network for Social Network analysis (www.insna.org) and its linked sites. The index of contents of the Social Networks journal was also searched. The online search was run on 5th January 2021 and later updated on 1st May 2022 to include papers published up to this date. The search strategy had no study design filters or restrictions to language as long as the paper could be found in English. Records were managed within a Mendeley library.

**Eligibility criteria.** The review included studies undertaken in any healthcare setting that reported the results of an SNA performed on networks among healthcare professionals (e.g., doctors, nurses, etc.) and other individuals involved in their professional networks (e.g., management, administrative support etc.). Examples of these networks include discussion networks, advice and knowledge sharing, and working on projects together. The healthcare setting was not restricted to a single geographical or organizational location, and could include wider interpersonal networks, such as the Parkinson network [13]. Veterinary or dental

professionals were not included. Studies of networks linking organizations, rather than individuals, were excluded. We excluded studies where network relations were defined solely by patient sharing, as this predicts person-to-person communication only in minority of instances [14].

We built upon Chambers et al.'s [8] classification method. We divided papers into three groups, which we termed level 1 to 3. Level 1 included studies reporting on the impact of an SNA-based intervention. Level 2 included studies describing existing social networks among healthcare professionals without reporting any follow-up action. Level 3 included descriptive studies that went on to suggest an SNA-based intervention intended to affect outcomes and behaviors. We added this additional category to shed light on the significant number of papers acknowledging the value of using SNA to inform the design of intervention plans, and the benefits associated with it.

**Study selection and data extraction.** Two Authors independently screened studies by title and disregarded those that they agreed to exclude. Studies where there was agreement for inclusion were independently screened by abstract by three Authors. Studies that appeared to meet the review inclusion criteria were forwarded to full-text evaluation and data extraction. The Cochrane EPOC (Effective Practice and Organisation of Care) Group criteria were used to assess the risk of bias by two Authors. Disagreements were discussed with a third Author.

## Results

The search returned 31,2867 unique papers, of which 102 met the eligibility criteria. Ten of these [15–24] were also included in Chambers et al.'s [8] review due to a crossover of search periods. We excluded these papers. The PRISMA diagram in Fig 1 below outlines the study selection process, and S1 Table outlines the number of records identified by database with a comparison to Chambers et al.'s [8] review. The comparison seems to suggest an increased use of social network approaches in healthcare studies over the past few years.

Four included studies met the level-1 [13, 25–27], 74 the level-2 [15–17, 19–21, 23, 28–94], and 24 the level-3 [18, 22, 24, 95–115] criteria.

Of the 102 papers, one third (n = 33) was conducted in the USA, 22 in Europe (excluding UK), 16 in low- and middle-income countries (LMIC), 11 in Australia, eight in the UK, seven in Canada, two in Japan, two in China and one in Malaysia. The Netherlands and Italy produced the largest number of papers in Europe. Compared to previous reviews mentioned earlier, we found an increased number of studies conducted in LMIC. The largest number of studies (n = 59) had participants from multidisciplinary teams, and were conducted in secondary care settings (n = 64). The number of participants ranged from 10 [71] to 16,171 [66]. The largest number of studies used surveys/questionnaires (n = 57), followed by direct observations (n = 7), mixed methods (n = 13), process logs or other administrative data (n = 9), interviews (n = 7), online platforms or forums (n = 5), and interaction data collected through sensors (n = 4).

We summarized the types of ties examined in the included papers into 10 categories to standardize the language (see S2 Table). We also grouped network measures into 36 categories (see S3 Table). These measures were used across studies to describe or analyze networks at the individual, dyadic, group, and whole network levels. We also created a distinct category for those papers performing only statistical analysis of network data, such as Exponential Random Graph Models (ERGMs), Multiple Quadratic Assignment Procedure (MQAP), and Stochastic Actor Oriented Models (SAOMs). Network visualization was included as a distinct category when it was the only social network method used.

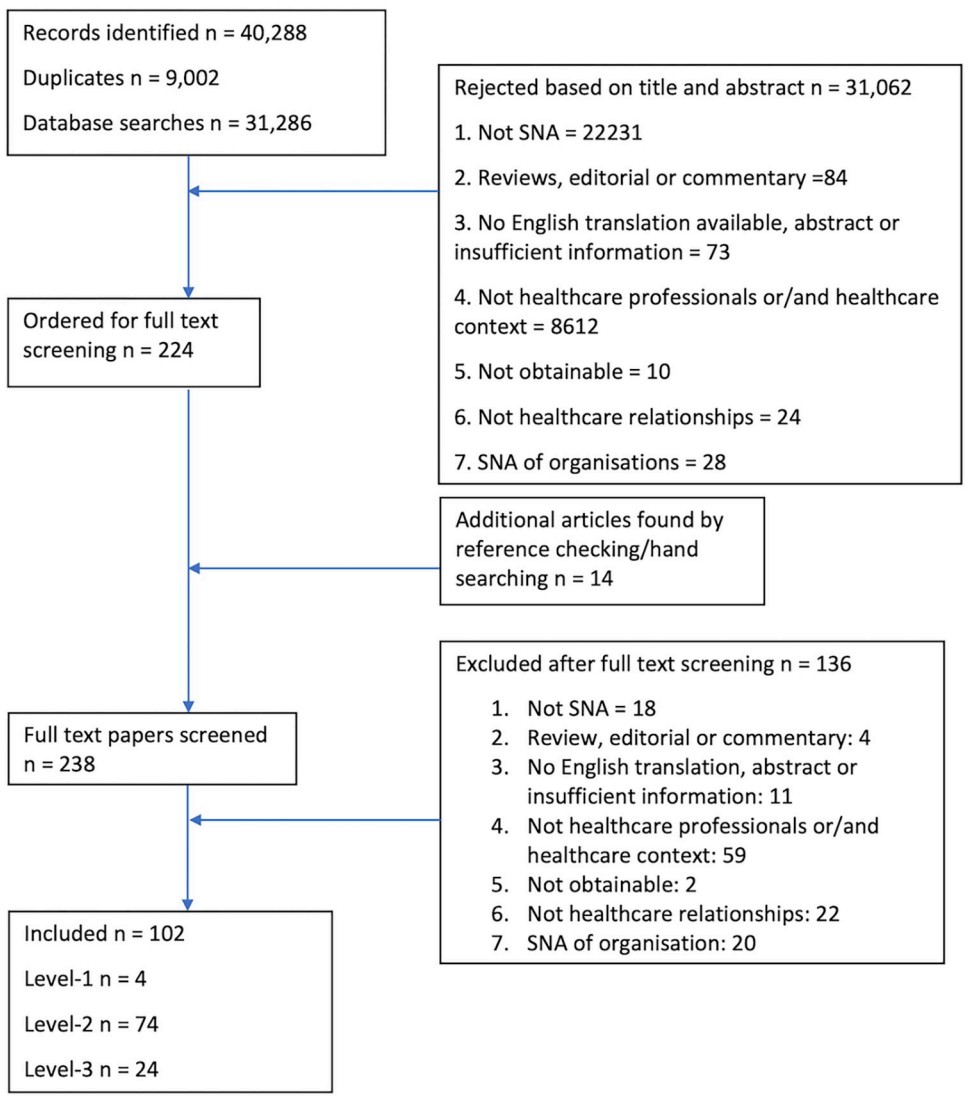

**Fig 1. Flow diagram of study selection process.**

## Level-1 studies

Table 1 below includes the level-1 studies, followed by a descriptive summary.

The four level-1 studies report on the results of SNA as part of an intervention, which we classified according to Valente [7]. Benton et al. [26] employed 'alteration' and 'induction' strategies by using shared project work to form new connections and increase interactions among network members. van der Eijk et al. [13] employed 'induction' through training events. The remaining two studies [25, 27] focused on 'individuals', by using social network methods to identify individuals who would act as champions. The impacts reported in the papers included structural network changes as well as changes in working practices and, in one study, staff safety outcomes. None of the studies reported on the impact on patient outcomes. The overall aim of the reported interventions was to improve organizational performance [26], patient care across the Parkinson's network [13], safe patient manual handling [25], and hand hygiene [27]. All four papers used the information from SNA to improve connectedness within the networks. A summary of the level-1 studies is provided in turn below.

**Table 1. The level-1 studies.**

| Ref | Country | Participants | Setting | Data collection | Type of tie | Network measure(s) | Key network findings | Network intervention | Network strategy | Recommendations |
|---|---|---|---|---|---|---|---|---|---|---|
| Benton 2015 [26] | Scotland, UK | 46 Nurse leaders | Secondary and community care | Survey | Communication | Density; average path length; network diameter | Information exchange network was mapped before and after an intervention bringing uncommected nurse leaders together to work on projects. Six months after there was an increase in network density and a reduction in average path length and more ties spanning different areas of work. Participants with low initial connectedness improved their number of ties. Connectedness and closeness improved considerably for those doing projects but not for individuals not involved in projects. | Results of SNA are fed back to participants. A subgroup of participants was then allocated to projects based on their interest in topics, and their low level of pre-existing connection. The projects required participants to communicate and work together to agree on actions to strengthen organizational strategy. | Alteration and induction | Use SNA followed by visual feedback to staff to stimulate positive change in the network. Bring disparate staff together in project teams to facilitate a sustained increase in connectiveness. Include staff with low baseline connectiveness. |
| van der Eijk 2015 [13] | Netherlands | 101 Multidisciplinary healthcare professionals involved with Parkinson care | Secondary and primary | Questionnaire and interview | Knowing each other; professional contacts | Number of connections; density; reciprocity | Participants completed a survey at baseline and one year after the training. Connections increased substantially in both networks from baseline to year one. Positive changes being associated with a central role of neurologists and nurse specialists committed to multidisciplinary care. Perceived team performance did not change. | Multidisciplinary professionals received training in technical and discipline-specific aspects of care, and in communication. Participants are granted access to an online community. There were semi-annual meetings and an annual national conference. | Induction | Provide shared training for multidisciplinary healthcare professionals treating a common disease. Invite participants to be involved in an online community. This may result in increased numbers of connections. |

*(Continued)*

**Table 1.** (Continued)

| Ref | Country | Participants | Setting | Data collection | Type of tie | Network measure(s) | Key network findings | Network intervention | Network strategy | Recommendations |
|---|---|---|---|---|---|---|---|---|---|---|
| Hurtado 2020 [25] | USA | 38 (pre) and 55 (post) nurses and nursing assistants | Community hospital | Survey | Advice | 7 Network centrality measures | Deployment of champions who had received technical and leadership training was associated with an increase in equipment use, safety compliance and incident reporting. There was a reduction in injuries to staff which was significantly different from 2 control sites. | SNA used to identify influencers in the area of safe patient handling. The top quintile nominations were invited to be champions. They were trained in technical aspects of patient handling and team leadership. They also participated in a number of quality improvement meetings. | Individuals | Use surveys to identify individuals who are already seen as 'go to' people for a particular topic. Identify champions, make them visible, and provide training in both technical skills and knowledge. Maintain leadership. Maintain connection with champions through regular meetings. |
| Lee 2019 [27] | Malaysia | 111 Health care workers | Secondary | Questionnaire | Communication | Geodesic distance; density; reciprocity; degree; closeness; betweenness | Hand hygiene compliance improved by a similarly degree in both peer and manager nominated champion arms. There was an improvement in hand hygiene practice and a preference for top-down leadership structure. | In one study arm staff nominated and ranked peers to become change agents. In the second arm, managers selected champions. Change agents and champions promoted hand hygiene in their local workplace. | Individuals | In order to change workplace behavior, select and train local champions; peer nominated or manager nominated champions have similar impact. |

Benton et al.'s [26] research was set in the National Health Service, Scotland. This was a quasi-experimental, pre-post intervention design. Analysis of the communication network of a group of nurse leaders was performed. Forty-six nurse participants from the acute and community setting participated to a baseline survey, which identified 18 participants for the intervention. Participants were selected because SNA data showed they were relatively weakly connected within the network. They were placed into one of three working groups based on their area of expressed expertise or interest. The aim was to influence the existing communication network by encouraging less connected participants to work together. To facilitate this, SNA data from the initial survey was fed back to all participants. The communication network was measured six months after the first data collection. Following involvement in the working groups, the selected 18 individuals showed substantial increase in number of ties. This was evidenced by a rise in connectedness score, which improved from 15.72 to 33.9, and closeness centrality which improved from 8.76 to 13.17. There were also improvement in global network efficiency and density, while the average path length reduced from 1.58 to 1.48. Network visualization showed more connections between professional groups. The Authors suggested that the wider network effects may have been affected by the feedback of the results of the first survey, which made people aware of their own position, and prompted curiosity about how they could change it. It also made people aware of the expertise available in peers. One weakness of the paper is that increase in connectedness among the 18 project participants was based on a survey done six months after the completion of the project groups. Hence, it is unclear whether the impact on network topology would be continued long term.

van der Eijk et al. [13] conducted a parallel group, mixed-methods study in the Netherlands. The study aimed to evaluate the Parkinson network, a nationwide organization with regional networks of health professionals. The study involved 101 multidisciplinary healthcare workers involved with Parkinson's care. Participants, who were based in hospital, nursing home or primary care settings, were selected to take part in a program on the basis of their location and 'motivation' (the latter term is not explicitly defined in the paper). They underwent a training course on multidisciplinary aspects of Parkinson's disease, and were given access to a database of expert therapists in their geographical location. There were also semi-annual meetings and an annual conference. Participants completed a survey on network connections and perceived team performance at baseline. One year later, a subsample was interviewed. There was a substantial increase in the number of 'knowing each other' connections from 1,431 to 2,175 ($p < 0.001$) and in 'professional contact' connections from 664 to 891 ($p < 0.001$). Neurologists and nurse specialists had a central position and were very well connected one year after the program implementation. Overall team performance did not change, but satisfaction with multidisciplinary collaboration increased significantly. There were no data on the impact of network characteristics on either patient outcome measures–such as symptom control or patient satisfaction, or process measures–such as rate of provision of evidence-based elements of care.

Hurtado et al. [25] used social network survey data to identify highly influential co-workers who were recruited as local champions in a safe patient handling education program. The Authors reported that previous studies in this context showed variable short- and long-term impact and that this may be due to a lack of proper methods for selecting workers best suited to exert influence. The study was carried out in critical care areas in one US hospital, and used a survey to collect data on advice seeking for safe patient handling. Individuals showing high centrality in the network were chosen as champions and were trained in safe handling. They were identified to other staff through announcements and wearing of ribbons. The results showed an increase in safety incident reporting, correct equipment use and safety compliance,

as well as reduction in staff injuries. Individual injury profile was significantly different from that of the two control hospitals in the same system.

Lee et al. [27] performed a parallel group study comparing two strategies to influence a behavior, hand hygiene compliance, through the use of local champions. The strategies were deployed on two similar medical wards. SNA showed there were few ties between the wards, suggesting that cross contamination was unlikely to occur. Staff on both wards were asked to nominate and rank peers in terms of their suitability to be hand hygiene champions. In one study arm, champions were selected on this basis. In the other study arm, managers selected champions without reference to the peer ranking. The champions themselves did not know how they had been selected. Trained observers used a validated approach to measure hand hygiene compliance during the study. Compliance increased substantially, from 48% to 66% in the peer selected champion arm, and from 50% to 65% in the manager selected champion arm. There was no statistical difference between the groups.

## Level-2 studies

Table 2 below includes the level-2 studies, followed by a descriptive summary.

Seventy-four studies were classified as Level-2. These are studies using SNA solely for descriptive or analytic purposes, without discussing about possible interventions aimed at changing or improving the structure or functioning of the networks. Twenty-three studies were from the USA, 14 from Europe (excluding UK), 14 from LMIC, seven from UK, nine from Australia, five from Canada and two from Japan. Forty-five studies used teams or mixed groups of healthcare professions as participants, 15 papers featured doctors only, 10 papers involved nurses, one study radiologists, one study psychologists, one also involved patients, and one had other types of healthcare professionals.

The majority of the studies (n = 46) were set in secondary care settings, followed by community (n = 9) and primary care settings (n = 5). Eight studies were conducted in mixed secondary and community, and primary and secondary settings. Finally, three studies were set in virtual settings, one in a university hospital, one in a cross sector and one in a nursing home. Twenty-six papers relied on surveys to collect network data, 17 used questionnaires, 10 used logs or administrative data, seven were based on mixed methods, six on observation, four on interviews, two on online platforms or forums, and two on interaction data from sensors.

Ten different types of ties were examined, the commonest being information and knowledge exchange. Nine papers described more than one tie [15, 34, 36, 44, 60, 64, 72, 75, 92]. Twenty-nine different network measures were used to describe the networks at the individual, dyadic, group and whole network levels. Statistical analysis was performed as the only analytical method in 10 studies. Burt et al. [42] is a theoretical paper suggesting different types of questions for name generators. Forty papers (60%) were published between 2010 and 2015, and thirty-four (40%) between 2016 to 1st May 2022.

## Level-3 studies

Table 3 below includes the level-3 studies, followed by a descriptive summary.

Twenty-four studies were classified as Level-3. Nine were conducted in the USA, seven in Europe (excluding UK), two in Australia, two in LMIC, two in Canada and two in China. Twelve used teams or mixed groups of healthcare professionals as participants, nine studies used doctors, two had other health professionals and one used healthcare providers and patients. The majority of the studies (n = 17) collected data in a secondary care setting, four in the community, one in primary care, one in primary and secondary care, and one in public health.

**Table 2. The level-2 studies.**

| Ref | Country | Participants | Setting | Data collection | Type of tie | Network measure (s) | Key network findings |
|---|---|---|---|---|---|---|---|
| Kim 2021 [91] | Korea | 222 Nursing students | University | Survey | Personal and social support | Indegree; outdegree; betweenness | A high level of subjective happiness is associated with a strong social network. Students with a high level of subjective happiness showed high network centrality. SNA can be used to improve nursing students' happiness by utilizing team-learning social networks within programs. |
| Haruta 2021 [90] | Japan | 52 Multidisciplinary healthcare workers | Secondary | Questionnaire | Advice | Clustering; density; degree; reciprocity; betweenness | Advice seeking network structures differed by topic areas. Nurses had highest centrality for all areas. The effect of feeding back the findings to healthcare professionals may have helped them to reflect on, and act upon their own networks. |
| Mukinda 2021 [92] | South Africa | 42 Managers and healthcare providers involved with maternal, newborn and child health | Primary and secondary | Questionnaire | Communication; social support | Degree; betweenness; density | Governance structures can support collaborative networks to improve cohesion between multidisciplinary teams by integrating missing links to improve information sharing and strengthen teamwork between frontline providers. |
| Bertoni 2022 [94] | Brazil | 133 Multidisciplinary or intensive care unit workers | Secondary | Questionnaire and interviews | Advice | In-degree; closeness; betweenness | Key players are not the same across the four ability-based networks. Thus, if responding, anticipating, learning, and monitoring are core activities that a resilient system displays, different individuals may take the lead on each of those roles. It is possible to investigate the contribution of individual players to resilience from a system perspective. |
| Smit 2021 [89] | Netherlands | 55 Multidisciplinary healthcare professionals | Primary | Survey | Collaboration | Degree; reciprocity | It is feasible to implement an interprofessional collaboration in practice (IPCP) program. Secondary data on the reporting of network metrics showed an increase in the number of contacts among the program participants. After the program, the program and non-program participants gained more collaborative, and diverse inter-professional networks. |
| Hayward 2021 [93] | Australia | 19 Multidisciplinary professionals involved in disability services | Cross sectors | Survey | Advice | Outdegree; indegree; betweenness | Nineteen individuals are identified who occupy positions of either boundary spanning (those linking people and groups) and/or opinion leadership (those that are sought for advice). Boundary spanners meet all criteria while opinion leaders do not. |

*(Continued)*

**Table 2.** (Continued)

| Ref | Country | Participants | Setting | Data collection | Type of tie | Network measure (s) | Key network findings |
|---|---|---|---|---|---|---|---|
| Durojaiye 2022 [88] | USA | 1647 Multidisciplinary pediatric trauma healthcare workers | Secondary | Electronic health records and interviews | Patient sharing | Network graph | Networks dealing with individual trauma cases are different between day and night. Network patterns for collaborative working are different during day versus night shifts. |
| Tasselli 2015 [85] | Netherlands | 118 Hospital professionals (65 nurses and 53 doctors) | Secondary | Survey | Knowledge transfer | Average degree centrality; hierarchy; average betweenness centrality | There are disciplinary cliques for knowledge transfer. Clinical directors facilitate knowledge transfer through their central network position. Junior doctors and nurse managers display both inter-professional and intra-professional centrality positions and are more likely to access valuable knowledge. |
| Wagter 2012 [61] | Netherlands | 108 ICU/MCU staff (senior doctors, nurses, residents and facilitating jobs) | Secondary | Questionnaire | Knowledge sharing | Densities; tie strength; reciprocity | ICU/MCU nurses formed cliques. There are unilaterally directed relations of senior doctors with nurses and patients. |
| Malik 2014 [35] | Pakistan | 48 Primary physicians and 5 district health administrators and line managers | Primary | Interviews and questionnaire | Advice seeking | Network graph | Primary physicians are aware of available expert knowledge, but advice-seeking behavior is dependent upon existence of informal social interaction with the senior specialists. |
| Patterson 2011 [28] | USA | 3 Emergency medical technician teams (EMT) (size of staff: N = 41; N = 67; N = 81) | Secondary | Administrative data | Familiarity (having worked previously together during shifts) | Number of partnerships; means; rates; proportions | On average, an EMT works with 19 different partners over the course of the year and there is significant variation in EMT partner familiarity across agencies. These patterns are considered an indicator of poor emergency medical services outcomes. |
| Groenen 2017 [55] | Netherlands | 214 Healthcare workers from 8 different professions | Secondary and community | Questionnaire | Patient-related contacts | Density; centrality | Almost all professionals in the network can reach other professionals in two steps. Only community-based midwives have connections with all other groups of professionals and represent 51% of all measured connections. The youth health doctors and nurses are mostly positioned on the edge, and are less connected. Obstetricians and community midwives have the highest score for betweenness centrality. |

(*Continued*)

**Table 2.** (Continued)

| Ref | Country | Participants | Setting | Data collection | Type of tie | Network measure(s) | Key network findings |
|---|---|---|---|---|---|---|---|
| Yuce 2014 [59] | Netherlands | 394 Hospital physicians | Secondary | Questionnaire | Advice | Density; average degree centrality | Advice seeking networks among doctors differ for medical and IT related issues. Trainees are just as likely to approach faculty on medical issues as peers, but more likely to approach peers on IT issues. Faculties go to peers for advice in medical practice, but not to trainees for technology-related advice due to the mentor system. Opinion leaders are different for the two domains. |
| Sibbald 2013 [50] | Canada | 6 Multidisciplinary healthcare teams from 2 primary health care team (PHCT) Practices | Primary | Questionnaire and interviews | Information exchange | Density; indegree | Respondents in the sample of PHCTs generally provide research information to only a few individuals on their teams and, overall, only a few individuals are providing the information. Key players in the knowledge uptake and dissemination process are residents, senior physicians, and nurse practitioners. |
| Benham-Hutchins 2010 [20] | USA | 25 Hospital staff and hand-overs (11 to 20 providers over 5 handoffs) | Secondary | Observation Snowball sampling | Communication | Betweenness; closeness; eigenvector; betweenness centralization; hierarchy | Each handoff network exhibits unique communication patterns and coordination. Most participants prefer verbal communication. |
| Burt 2012 [42] | USA | 25 Hospital physicians at quality improvement sites | Secondary | Survey | Different types of ties and name generator questions | Comparison of name generators | Some physicians maintain a social network organized around a specific colleague who perform multiple roles, while others maintain highly differentiated networks. A set of 5 of the 8 name generators used is needed to distinguish the networks of these physicians. Multiple survey questions are needed to elucidate networks of knowledge sharing among physicians. |
| Shokoohi 2013 [73] | Iran | 140 Students (70 clerks, 45 interns and 25 residents) in an educational hospital | Secondary | Questionnaire | Knowledge transfer | Density; indegree; outdegree; reciprocity | Residents are consulted with almost as same as attends on diabetic foot ulcers, hence showing a prominent role in knowledge transfer. The density of clerks-residents and interns-residents is higher than clerks-attends and interns-attends. Indegree centralization in attends-related networks is greater than residents-related networks. |

(*Continued*)

**Table 2.** (Continued)

| Ref | Country | Participants | Setting | Data collection | Type of tie | Network measure (s) | Key network findings |
|-----|---------|--------------|---------|-----------------|-------------|---------------------|----------------------|
| Fuller 2012 [38] | Australia | Two case studies of chronic illness service partnerships (42 partnership staff and 19 informants) in 2 Australian sites | Community | Survey | Communication | Degree; betweenness | Participants in both research groups considered that the network survey accurately described the links between workers related to the exchange of clinical and cultural information, team care relationships, involvement in service management, planning and policy development. Aboriginal workers have a high number of direct links in the exchange of cultural information–suggesting a role of cultural resource–but have fewer direct links in the exchange of clinical information and team care. |
| Patterson 2013 [39] | USA | 103 Clinicians and non-clinician staff in a multidisciplinary Emergency Department (ED) team | Secondary | Survey | Communication | Density; centralization; indegree | There is wide variation in the magnitude of communication cohesion (density) and concentration of communication between clinicians (centralization) by day/night shift and over time. There is also variation in indegree centrality (a measure of power/influence) by day/night shift and over time. |
| Venkatesh 2011 [86] | USA | 1,120 Hospital physicians and other staff (doctors, paraprofessionals, administrative personnel) | Secondary | Survey | Advice | Degree | Ingroup and outgroup ties play a critical role in influencing e-healthcare system use. Further, such use has a positive effect on a variety of quality-of-care metrics that in turn influence patient satisfaction. |
| Barth 2015 [53] | UK | Pediatric surgery team in 40 pediatric cardiac surgical procedures | Secondary | Observation | Communication | Degree centralization; density; closeness centralization; betweenness centralization; reciprocity | In complex surgical procedures, communication patterns are more decentralized and flatter. In critical transition phases of the procedure, communication is characterized by higher information sharing and participation. |
| Tsang 2012 [30] | Taiwan | 60 Nurses in a dialysis department of a medical centre | Secondary | Survey | Work-related information exchange | Degree; closeness; betweenness | Organizational citizenship behavior (OCB) increases with centrality in both work and friendship networks. Experienced nurses show high centrality in the work networks. In the friendship network, those with high centrality are not necessarily of higher rank in the organization. OCB induced by social ties is satisfactory. It directly increases work satisfaction and alleviates work stress. |

(*Continued*)

**Table 2.** (Continued)

| Ref | Country | Participants | Setting | Data collection | Type of tie | Network measure (s) | Key network findings |
|---|---|---|---|---|---|---|---|
| Tavakoli Taba 2016 [64] | Australia | 31 Breast imaging radiologists | Secondary | Survey | Professional interaction and knowledge sharing | Degree; density; effective size; efficiency; constraint; hierarchy; mean tie strength | There is a positive relationship between diagnostic performance and degree centrality and network size, but a negative relationship with constraint and hierarchy. Overall, the results suggest that radiologists interacting with a closely knit cluster through multiple primary ties–resulting in higher constraints for them–performed worse than radiologists with effective, less constrained (or non-redundant) contacts. |
| Walton 2010 [21] | Canada | 6 Teams in a pediatric ward (doctors, residents and medical students) | Secondary | Observation and questionnaire | Patterns of team interaction | Betweenness | Three different patterns of verbal interaction are observed. In most cases, the attending physician are most talkative and many students and residents spoke infrequently. |
| Paul 2014 [32] | USA | 33 Primary physicians | Community | Survey | Knowledge sharing | Reciprocity; triadic dependence | A physician influential discussion, and a patient-sharing networks are analyzed. Patterns of influential discussions among physicians exhibit triadic dependence. Reduction in reciprocity due to triadic and other higher-order forms of clustering. Geographically proximal physicians are more likely to share patients. |
| Tighe 2012 [63] | USA | 55 Members of Anesthesiology department and 29 patients | Secondary | Service schedule | Communication | Various measures for size and structure of the network, and information flow are used. Many node-level measures are also used | The network exhibits a relatively low density and clustering coefficient, suggesting a low level of redundancy. The high Krackhardt hierarchy score suggests multiple levels of responsibility and supervision between attending, fellow, and resident anesthesiologists. Despite the relatively small size of the core regional anesthesia and perioperative pain medicine) team, its interactions with a large number of services over multiple geographic locations lead to considerable network complexity. |
| Hinami 2019 [31] | USA | 2280 Prescribers of opioid analgesic | Secondary and community | Prescription claim data | Shared benefactors | K-shell centrality | SNA identifies two small, interconnected prescriber communities of high-volume pain management specialists, and three sparsely connected groups of predominantly low-volume primary or emergency medicine clinicians. The sparsely connected clinicians are a risk factor for uncoordinated opioid prescribing. |

(*Continued*)

**Table 2.** (Continued)

| Ref | Country | Participants | Setting | Data collection | Type of tie | Network measure (s) | Key network findings |
|---|---|---|---|---|---|---|---|
| Long 2014 [52] | Australia | 68 Cancer research networks of hospital-based clinicians and university-based researchers | Secondary | Online Survey | Collaboration | Density; components; External-Internal (E-I) index; clustering coefficient | Geographic proximity and past working relationships have significant effects on the choice of current research collaboration partners. Future intended collaborations include a significant number of weak ties and ties based on other members' reputations. |
| Dauvrin 2017 [72] | Belgium | 575 Healthcare professionals working in inpatient and outpatient services | Secondary | Survey | Professional relationships | Degree | At the dyadic level, no significant associations are found between ego cultural competence and alter cultural competence, except for subjective exposure to intercultural situations. No significant associations between centrality and cultural competence, except for subjective exposure to intercultural situations: The most central healthcare professionals are not more culturally competent than less central health professionals. |
| Altalib 2019 [37] | USA | 66 Epilepsy care facilities and 165 providers | Secondary and primary | Secondary data and interviews | Patient sharing | Degree; betweenness; closeness | Across Veterans Affairs Healthcare System (VA) facilities, neurologists are found to be higher on average node degree, betweenness, and closeness centrality measured followed by mental health professionals, then primary. Providers, across disciplines, have higher centrality measures in Epilepsy Centres of Excellence (ECOE) hubs compared to spoke referral facilities and non-affiliated networks. Facilities had a variety of network configurations. |
| Stewart 2012 [57] | Thailand | 46 Pediatric pain practitioners | Secondary | Online discussion forum | Knowledge sharing | Degree; closeness; betweenness; coreness | The network is dominated by one institution and a single profession. There is also evidence of a varied relationship between reading and posting content to the discussion forum. SNA reveals a network with strong communication patterns and users who are central to facilitating communication. SNA also reveals that there is a strong interprofessional and interregional communication, but a dearth of non-nurse participants are identified as a shortcoming. |

(*Continued*)

**Table 2.** (Continued)

| Ref | Country | Participants | Setting | Data collection | Type of tie | Network measure (s) | Key network findings |
|-----|---------|--------------|---------|-----------------|-------------|---------------------|----------------------|
| Blanchet 2013 [62] | Ghana | 12 Ghanaian districts; 53 individuals (hospital managers, nurses and district/regional/ national health officers, district education officers, community health volunteers, coordinators) | Secondary | Interviews | Coordination and collaboration | Density; distance; degree; betweenness | The departure of an international organization, caused a big shock to the health system, resulting in a change in relationships and power structures within the network. The system shifts from a centralized and dense hierarchical network, to an enclaved network made up of five sub-networks. The sub-networks are less able to respond to shock, circulate information and knowledge across scales or implement solutions. The network is less resilient, yet it responds better to management's need to access information. |
| Alexander 2015 [70] | USA | 12 Certified nursing assistants and registered nurses | Nursing home | Observation | Communication | Network graph | Direct interaction between nurses is higher in the low IT sophistication home and occur in more centralized locations compared to the high IT sophistication home. |
| Laapotti 2016 [71] | Finland | 10 Healthcare professionals with managerial roles, a chair and a secretary | Secondary | Observation | Interactions between team members | Network graphs | The structure of the interaction network reveals that interactions reflect the organizational roles of the participants, as they are focused on the chair. |
| Lai 2020 [79] | Taiwan | 50 Nurses of surgical wards | Secondary | Questionnaire | Friendship | Network graphs; regression | Perceived usefulness, perceived ease of use, and social influence affect behavioral intention to use cloud sphygmomanometer. Besides, perceived ease of use and social influence positively influence perceived usefulness of cloud sphygmomanometer. Peers are helpful in motivating medical staff to use the cloud sphygmomanometer. |
| Mascia 2014 [40] | Italy | 104 Primary physicians and pediatricians | Primary and Secondary | Questionnaire | Knowledge exchange | Outdegree | The number of relationships with hospital colleagues is associated with use of evidence-based medicine. |
| Shafiei 2018 [82] | Iran | 64 Nurses | Secondary | Interviews | Work-related interactions | Degree; closeness; betweenness; eigenvector | Interactions within a department are strong but those between nurses of different departments are not. |
| Kawamoto 2020 [76] | Japan | 76 Intensive Care Unit (ICU) healthcare professionals (HCP) | Secondary | Wearable sensors | Face-to-face interactions | Degree; betweenness; eigenvector | Wearable sociometric sensor badges show nurses have a pivotal role in communication amongst the ICU HCP. |

(*Continued*)

**Table 2.** (Continued)

| Ref | Country | Participants | Setting | Data collection | Type of tie | Network measure (s) | Key network findings |
|---|---|---|---|---|---|---|---|
| Cavalcante 2018 [77] | Brazil | 3 Healthcare professionals (1 doctor and 2 nurses) and their networks' members (19 people) | Mobile Urgent Care Service | Interviews | Work-related interactions | Size; Density | The networks consist of (mutual) relationships that satisfy the demands and needs of service users in an integrated manner while attempting to respect the knowledge and autonomy of each member. Nevertheless, the networks are characterized by poor collaboration ("star" shape) with few transposition points (bridges). This leads to problems in the performance of tasks and mental suffering at work. |
| Lazzari 2019 [51] | UK | 42 Dementia professionals in 3 teams, and 42 patients with Alzheimer's disease | Secondary | Observation | 2-mode networks of professionals by services provided | Degree | All professional roles are involved in the case of patients' biological and sociologic personhood. The nurse is the most central figure in the case of biological personhood. |
| Currie 2012 [87] | UK | 36 Pediatric nephrology multidisciplinary teams | Secondary and community | Survey | Knowledge exchange | Degree; betweenness; brokerage roles; density | Knowledge-brokering roles are influenced by professional hierarchy, particularly in the case of clinical knowledge and even more so with medical knowledge. |
| Chung 2014 [84] | Australia | 107 General Practitioners | Primary | Questionnaire | Advice | Density; inclusiveness; components | Considering the GP-patient encounter as a complex system, the interactions between the GP and their personal network of peers give rise to "aggregate complexity," which in turn influences the GP's decisions about patient treatment. GPs in simple profiles (i.e. with low components and interactions) in contrast to those in nonsimple profiles, indicate a higher responsibility for the decisions they make in medical care. |
| Yuan 2020 [49] | USA | 207 Nurses in 6 clinical units in an academic hospital | Secondary | Survey | Advice | Mean peer belief (ego-network analysis) | Although mean beliefs across the entire peer network have no effect on individuals' system use, shared peer beliefs were associated with nurses' increased use of the IT system. Reinforcement by the social network appears to influence whether individuals' own beliefs translate into system use, providing further empirical support that social networks play an important role in the implementation of health information technology. |

*(Continued)*

**Table 2.** (Continued)

| Ref | Country | Participants | Setting | Data collection | Type of tie | Network measure(s) | Key network findings |
|---|---|---|---|---|---|---|---|
| Uddin 2013 [69] | Australia | 85 Physicians networks | Secondary | Health insurance claim dataset | Collaboration | Density Degree; betweenness centralization Exponential random graph models (ERGMs) | Collaboration structures among physicians affect hospitalization cost and hospital readmission rate. |
| Benton 2014 [48] | Scotland, UK | 27 Senior nurses | Virtual | Survey | Communication | Degree; betweenness; eigenvector; density; average path length; network diameter | The majority of nurse leader who participated in the Global Nursing Leadership Institute 2013 Programme are poorly connected in social media, i.e., they have low indegree and outdegree scores. Existing connections are centered on geographic proximity and participation in regional and global bodies. |
| Mundt 2015 [54] | USA | 155 Primary health care professionals from 31 teams at 6 primary care clinics | Community | Survey | Communication | Density; centralization | Teams with dense interactions are associated with fewer hospital days and lower medical care costs. Conversely, teams with interactions revolving around a few central individuals are associated with increased hospital days and greater costs. |
| Quinlan 2013 [74] | Canada | 49 Nurse Practitioners in primary healthcare teams | Primary | Survey | Knowledge transfer | Within-team reciprocation; within-team degree centrality | Mutual understanding increases from one clinical decision to another in some teams and decreases in others. The new Nurse Practitioners play a crucial role in facilitating mutual understanding and knowledge exchange in the newly created multidisciplinary teams. A well-functioning team has effective intrateam knowledge exchange. |
| Li 2016 [58] | Netherlands | 621 Healthcare Professionals (users) and 723 threads over 40 forums | Virtual | Online discussion forum | 2-mode network of forum users by discussion threads | Density; centralization; diameter; average path length; SAOMs | The participation level in the discussion within the online community is low in general. A change of lead contributor results in a change in learning interaction and network structure. Health professionals are reluctant to share knowledge and collaborate in groups, but are interested in building personal learning networks or simply seeking information. |
| Sullivan 2019 [65] | UK | 39 Trainee doctors in an acute medical unit | Secondary | Survey | Advice | Degree; betweenness; density | Information and influence relating to different aspects of practice have different patterns of spread within teams of trainee doctors. Influencers in clinical teams have particular characteristics, and this knowledge could guide leaders and teachers. |

*(Continued)*

**Table 2.** (Continued)

| Ref | Country | Participants | Setting | Data collection | Type of tie | Network measure (s) | Key network findings |
|---|---|---|---|---|---|---|---|
| Zappa 2011 [16] | Italy | 711 Physicians | Secondary | Survey/ Questionnaire | Knowledge sharing | ERGMs | Knowledge flows informally in mutual information-seeking relationships. Physicians tend to cluster in small groups of proximate and similar peers. The propensity to share knowledge is affected by individual-specific characteristics. |
| Aylward 2012 [47] | USA | 286 Pediatric Psychologists | Primary and Secondary | Survey | Mentoring | Density; indegree; outdegree; closeness; betweenness; average geodesic distance | The field of pediatric psychology is interconnected with professionals learning from multiple mentors in multiple settings. The average "degrees of separation" between individuals in the network is 5.30. |
| Benammi 2019 [78] | Morocco | 58 Members of an Acute Care Unit (ACU) in a university hospital | Secondary | Survey | Communication | Density; degree and betweenness centralization; degree and betweenness centrality | ACU network shows a moderate degree centralization, and lower betweenness centralization. The team is connected by well-positioned members to support inter-team communication, and is dominated by a number of gatekeepers, with low degree of communication among different function team members. |
| Bachand 2018 [66] | USA | 8338 Women with breast cancer in 157 physician peer groups (made up of 16,171 physicians) | Secondary | Surveillance, epidemiology, and end results-Medicare data | Patient-sharing | Ingroup density; transitivity | Surgical delays vary substantially across physician peer groups, and are associated with provider density and patient racial composition. Women in physician peer groups with the highest provider density are less likely to receive delayed surgery. |
| Bae 2017 [33] | England, UK | 54 Nurses in an acute care hospital unit | Secondary | Survey | Mutual support | Degree; closeness; betweenness; eigenvector density; shortest path; reciprocity; transitivity | Providers of mutual support claim to give their peers more help than these peers gave them credit for. Those who work overtime provide more mutual support. |
| Fong 2017 [43] | Taiwan | 100 Multidisciplinary staff members in 3 Intensive Care Unit (ICU) in an academic teaching hospital | Secondary | Questionnaire | Communication | Cluster analysis (k-means) | Distinct patterns and categories of influencers (well-rounded, relational, and knowledge-based) are identified using a clustering approach. Knowledge of how influence is distributed across the care team could lead to a better planning of change initiatives. |
| Creswick 2010 [19] | Australia | 45 Health professionals in a renal ward | Secondary | Questionnaire | Advice seeking | Geodesic distance; density; average strength of ties; reciprocity; degree; betweenness | On average, there is little interaction between each of the staff members in the medication advice-seeking network, with even less interaction between staff from different professional groups. Nurses are mainly located on one side of the network and doctors on the other. However, the pharmacist is quite central in the medication advice seeking network as are some senior nurses and a junior doctor. |

(*Continued*)

**Table 2.** (Continued)

| Ref | Country | Participants | Setting | Data collection | Type of tie | Network measure (s) | Key network findings |
|---|---|---|---|---|---|---|---|
| Dauvrin 2015 [60] | Belgium | 507 Healthcare professionals | Secondary and primary | Questionnaire | Problem-solving, advice-seeking, and socialization | Indegree | Cultural competence of the healthcare staff is associated with the cultural competence of the leaders. The leadership effect varied with the degree of cultural competence of the leaders. |
| Wong 2015 [36] | USA | 98 Pediatric Intensive Care Unit staff | Secondary | Survey | Information seeking, social influence and social support | Degree; density | Amongst the 3 networks, there are no weakly connected groups. Few individuals report no links to a colleague. The number of links among colleagues is greatest for the information seeking network, followed by social influence, and social support. Five individuals, three of whom have formal leadership roles, are amongst the 10 most influential team members in all 3 networks. |
| Hurtado 2018 [67] | USA | 38 Patient care workers | Community hospital | Survey | Advice seeking | Degree; reciprocity | There is a positive correlation between identifying more peers for safe patient handling advice and using equipment more frequently. Nurses with more reciprocal advice seeking nominations use safe patient handling equipment more frequently. However, nurses consulted more do not use equipment more frequently than nurses with fewer nominations. |
| van Beek 2013 [29] | Netherlands | 391 Nursing staff from 37 long-term care dementia units | Community | Questionnaire | Communication | In-group density | In units with more networks between nursing staff and relatives of residents, staff treated residents with more respect and were more at ease with residents. Social networks were also positively related to staff's organizational identification which, in turn, related to their work motivation and their behavior towards residents. |
| Boyer 2010 [23] | France | 104 Healthcare professionals in a hospital | Secondary | Questionnaire | Information sharing | Ingroup centrality; prestige; clique indicators | Centrality, prestige and clique indicators are highly correlated. Physicians have the highest scores for the three indicators. Older age is found to be associated with higher centrality and clique scores. |
| Anderson 2011 [56] | USA | Operating room staff (n = 733 interdisciplinary members) of 2 surgical specialties | Secondary | Staffing data on surgical cases in the 29 operating rooms | Individual affiliation to surgical cases | Degree; closeness; betweenness; eigenvector; core/periphery | Both surgical services show a core/periphery network structure. Team coreness is associated with the length of the case. Procedure start time predicts the team coreness measure, with cases starting later in the day less likely to be staffed with a high core team. Registered nurses constitute the majority of core interdisciplinary team members in both groups. |

(*Continued*)

**Table 2.** (Continued)

| Ref | Country | Participants | Setting | Data collection | Type of tie | Network measure (s) | Key network findings |
|---|---|---|---|---|---|---|---|
| Brewer 2020 [75] | USA | 268 Nursing staff in 24 Patient Care Unit (PCUs) | Secondary | Web-based questionnaire | Information sharing and advice seeking | Average distance; betweenness; clique count; clustering; density; diffusion; eigenvector; fragmentation; hierarchy; isolates; size; degree | In clinical workplaces with high day-to-day staff variation, several network characteristics remain stable over time. Hierarchy, fragmentation and cliques are unstable. |
| Lower 2010 [15] | Australia | 13 Multi-disciplinary teams in hearing services | Community | Questionnaire and interviews | Information exchange; referrals; working relationships | Degree; average number of ties | Nurse audiometrists, WorkCover and agricultural retailers have the lead role in disseminating information on hearing health within the network. For client referrals the nurse audiometrists, private audiometry services, general practitioners, ear, nose and throat specialists and industry groups play the major roles. |
| Quinlan 2010 [17] | Canada | 29 Nurse practitioners in primary-care teams | Community | Survey | Mutual understanding | Within-team density; flow-betweenness centralization | In two teams mutual understanding increases with time. In the other two teams, it decreases. As the overall mutual understanding within the team decreases, the facilitation of mutual understanding becomes more centralized among few team members; conversely, as mutual understanding increases, the facilitation becomes more equally distributed. The inverse relationship exists in all teams, except in team. |
| Edge 2019 [45] | UK | 138 Foundation doctors in one NHS trust | Secondary | Observational study | Physical contact | Degree; density; density by groups; assortativity | Direct network links to vaccinated colleagues increase an individual's likelihood of being vaccinated. |
| Espinoza 2018 [44] | Chile | 53 Inter-professional teams (409 professionals) at a university hospital | Secondary | Questionnaire and interview | Advice and personal support | Density; isolates; centrality; Within-group cohesion | For the work advice network, when a team structures itself around one professional, this allows its members to approach and be approached easily and facilitates information exchange. Teams with the least satisfaction reveal a fragmented structure with members organized as subgroups. The organization of social support networks is even more fragmented, with half of them being isolated from the rest of the team. |
| Crockett 2018 [80] | Canada | 22 Healthcare professionals in 18 general Emergency Departments | Secondary | Interviews | Information seeking | Content analysis | Health care professionals sought information both formally and informally, by using guidelines, talking to colleagues, and attending pediatric related training sessions. Network structure and processes were found to increase connections, support practice change, and promote standards of care. |

(*Continued*)

**Table 2.** (Continued)

| Ref | Country | Participants | Setting | Data collection | Type of tie | Network measure (s) | Key network findings |
|---|---|---|---|---|---|---|---|
| Pomare 2019 [34] | Australia | 23 and 27 Clinical and non-clinical staff members in 2 headspace centres | Youth mental health service | Survey | Collaboration, advice, problem solving | Degree; sub-group cohesion; density; centralization | Staff of headspace (clinical and non-clinical) show a tendency to collaborate with colleagues outside of their professional group, compared to within. Networks are well connected when staff collaborate in routine work and when faced with uncertainty in decision-making. There are fewer interactions during times of role uncertainty. The headspace centre that had been in operation for longer show greater indicators of cohesiveness. |
| Choudhury 2018 [81] | USA | 3 Large-sized integrated delivery networks; 14 hospitals; 288 physicians; 353 prescriptions | Secondary | Medical prescriptions and affiliations datasets | Affiliation | Diffusion models | Physicians affiliated to same hospital and integrated delivery network contribute highly in the diffusion process. The weighted edge approach is better able to explain diffusion of influence in terms of prescribing patterns. |
| Palazzolo 2011 [68] | USA | 3 Multidivisional healthcare teams (n = 126 individuals) in 1 hospital | Secondary | Email archives | Communication | Betweenness; contribution index; group betweenness; core/periphery; density; structural holes; connectivity | SNA of email communications of three teams caring for patients with different complex long-term conditions reveal distinct patterns and structures. Team metrics varied over time. Teams' network characteristics may explain their functioning. |
| Hornbeck 2012 [46] | USA | Healthcare workers (HCW) and patients in 1 Medical intensive care unit | Secondary | Mote-based sensor network | Physical contact | Agent-based simulation | Electronic sensor derived data on HCW interactions with other HCW's and patients reveal that a small number of HCWs were responsible for a large number of interactions. |
| Shoham 2015 [41] | USA | 69 Co-workers listed by 48 clinical team members in a burn intensive care unit | Secondary | Questionnaire | Communication | Degree; betweenness; density | The analysis revealed three distinct sets of team members caring for two sets of patients. The five clinical team members most central to the network included three physicians, a social worker, and a dietitian. |
| Zappa 2014 [83] | Italy | 106 Oncologists | Virtual community | Emails | Cooperation | SAOMs | Emergent network effectively represented by a small number of local rules, i.e., actors' behaviors of counterpart's selection in their neighborhood. |

Ten studies used questionnaires to collect data, three relied on mixed methods, four used surveys, three interviews, two collected interaction data from sensors, one used direct observation, and one an online platform or forum. Seven different types of ties were analyzed across studies. Two studies analyzed more than one tie [108, 113, 114]. Twelve different network measures were used to describe or analyze networks at the individual, dyadic, group and whole network levels. Statistical analysis relying on ERGMs and MRQAP were used five times.

**Table 3. The level-3 studies.**

| Ref | Country | Participants | Setting | Data collection | Type of tie | Network measure(s) | Key network findings | Recommendations | Network strategy |
|---|---|---|---|---|---|---|---|---|---|
| Xu 2021 [115] | China | 5247 Healthcare Workers | Secondary | Survey | Discussion | Density; degree | A vaccination consulting network of 1817 members is reconstructed. The network shows low density. Twenty-two influential members are identified. Lack of discussion is associated with vaccine hesitancy. Department leads are particularly influential as promoters of vaccination. | Use influential individuals as role models to encourage vaccine uptake. | Individuals |
| Jippes 2010 [18] | Netherlands | 81 Gynecologists and pediatricians and 63 residents in O&G and Pediatrics | Secondary | Questionnaire and interviews | Communication | Degree; closeness; betweenness | Social connections are more important than training for uptake of a new practice. A strong association is found between closeness centrality and adoptive behavior, and a moderate effect of degree centrality. | Incorporate individuals who have both strong and weak ties in 'teach-the-teacher' courses. | Individuals |
| Mascia 2018 [111] | Italy | 97 Pediatricians in 2 Local Health Authorities (LHAs) | Community | Questionnaire | Advice | ERGMs | In both LHAs, physicians tend to reciprocate advice ties; there is considerable clustering in advice-seeking. | Create new opportunities for knowledge exchange, such as taskforces or training programs. | Induction |
| Llupià 2016 [97] | Spain | 235 Healthcare workers in 1 hospital | Secondary | Interviews | Information exchange | ERGMs | Similarity in vaccination behavior does not play a significant role in the probability of being connected to another healthcare worker. | Use SNA to guide the design, implementation, evaluation of a health promotion campaign. For example, messages could be tailored by professional category or strategy could be implemented to foster communication among different professional categories. | Segmentation and induction |

*(Continued)*

**Table 3.** (Continued)

| Ref | Country | Participants | Setting | Data collection | Type of tie | Network measure(s) | Key network findings | Recommendations | Network strategy |
|---|---|---|---|---|---|---|---|---|---|
| Meltzer 2010 [102] | USA | 56 Physicians attending on the general medical services in 1 hospital | Secondary | Questionnaire | Communication | Degree; Net degree of team; betweenness; density | Connections of team members outside the team are important for dissemination of information or influence. Connections of team members inside the team are important for within-team coordination, knowledge sharing and communication. | Use SNA to decide whom to select for a quality improvement team, and how to structure the team. When influence through direct social interaction is important, choose individuals who can reach the largest number of persons outside the team. The use of degree alone to select team members may produce many redundant ties. | Individuals |
| Polgreen 2010 [103] | USA | 148 Multidisciplinary healthcare workers in 1 hospital | Secondary | Observational data and simulation | Physical contact | Number of contacts | Preferentially vaccinating healthcare workers in more connected job categories yield a lower attack rate and fewer infections in a simulation. | Identifying workers with many contacts might aid targeting vaccinations to optimize impact on flu spread. | Individuals |
| Mascia 2011 [24] | Italy | 297 Hospital physicians in 6 hospitals | Secondary | Questionnaire | Advice | MRQAP | Physicians reporting similar attitudes toward evidence-based medicine (EBM) are more likely to exchange information and advice. | Foster heterophily when multidisciplinary cooperation is required. Identify groups exhibiting desired attitudes and behaviors. Adopt organizational arrangements, processes and informal meetings to foster collaboration. | Induction and segmentation |
| Sykes 2011 [22] | USA | 151 Hospital physician in 1 hospital | Secondary | Survey | Advice | First degree centrality; second degree centrality | Both first-degree (direct) and second-degree (indirect) centrality negatively influence electronic medical records (EMR) system use. Physicians with more connections are less likely to be early users of EMR. | Be aware that resistance to EMR systems is greater among physicians with high centrality who should then be the target of resources to reduce such resistance. | Individuals |

(Continued)

**Table 3.** (Continued)

| Ref | Country | Participants | Setting | Data collection | Type of tie | Network measure(s) | Key network findings | Recommendations | Network strategy |
|-----|---------|--------------|---------|-----------------|-------------|--------------------|--------------------|-----------------|------------------|
| Pinelli 2015 [105] | USA | 72 Multidisciplinary healthcare professionals in 1 hospital | Secondary | Interviews | Communication | Size; density; strength of tie; betweenness | Most communication is synchronous. Most communication events occur between the primary nurse and the patient, and the care coordinator and primary nurse. | Improvements in discharges are possible by reorganizing systems to optimize communication. SNA could offer a cost-effective way to improve patient care provision. | Alteration |
| Mascia 2015 [107] | Italy | 297 Hospital physicians in a Local Health Authority (LHA) | Secondary | Questionnaire | Information exchange | MRQAP | Institutional and professional homophily affect inter-physician networks. Professional homophily is more relevant than institutional affiliation for collaborative ties. | Foster collaboration across heterogeneous groups of physicians from different specializations. | Induction |
| Shoham 2016 [99] | USA | 71 Multidisciplinary healthcare professionals in a hospital burn unit | Secondary | Survey | Discussion | Density; degree; ERGMs | Members of all roles are involved in a higher percentage of inter- than intra-professional ties. Physicians are most central to the network. Nurses are significantly more likely to connect with other nurses. | Consider purposefully developing the role of nurses within the team. | Segmentation |
| Gorley 2016 [106] | Canada | 227 Participants in a BC Sepsis network | Secondary | Questionnaire and interviews | Knowledge sharing | Density; Centrality | Eleven participants stand out as hubs (high degree centrality). These individuals have many connections with people who trust them. | When launching a new network or strengthening an existing network for quality improvement, several recommendations are offered (e.g., to seek and include distributed leaders in the network). | Individuals and Induction |

(*Continued*)

**Table 3.** (Continued)

| Ref | Country | Participants | Setting | Data collection | Type of tie | Network measure(s) | Key network findings | Recommendations | Network strategy |
|---|---|---|---|---|---|---|---|---|---|
| Mascia 2013 [100] | Italy | 297 Hospital physicians in 6 hospitals; 1 Local health unit | Secondary | Questionnaire | Advice | Coreness; network authority | The overall network shows a core-periphery structure. There is a negative association between physicians' attitudes toward evidence-based medicine (EBM) and the coreness they exhibited in the professional network. Network centrality indicators confirm a negative association between physicians' propensity to use EBM and their structural importance in the professional network. | Policy makers can foster collaboration across staff with different propensities to use EBM by relying on organizational arrangements, informal meetings, and use of medical leaders to persuade other professionals to collaborate more with EBM user. | Individuals and Induction |
| Creswick 2015 [101] | Australia | 101 Hospital staff members in 1 teaching hospital | Secondary | Questionnaire | Advice | Density; reciprocation; indegree | Medication advice-seeking networks among staff on hospital wards are sparse, information sharing across professional groups is modest, and rates of reciprocation of advice is low. Senior physicians are weakly integrated into medication advice networks; pharmacists and junior physicians play central roles. | Policies to advance the advice-giving networks between senior and junior physicians may improve medication safety as one ward with stronger networks had lower prescribing error rate. | Segmentation |
| Marques-Sanchez 2018 [98] | Spain | 196 Multidisciplinary healthcare professionals | Secondary and primary | Questionnaire | Internal and external advice | Outdegree (internal and external ties) | For physicians, external ties improve the performance at an individual and team level, yet external ties are not relevant for nurses' work performance. | Use SNA to facilitate healthcare professionals sharing information within and across organizations. | Alteration |

(*Continued*)

**Table 3.** (Continued)

| Ref | Country | Participants | Setting | Data collection | Type of tie | Network measure(s) | Key network findings | Recommendations | Network strategy |
|---|---|---|---|---|---|---|---|---|---|
| Kothari 2014 [114] | USA | 13 Public health practitioners | Community | Questionnaire and interviews | Interaction, support, and professional relationships | Cliques; degree; closeness; betweenness | Participants' report on their experience with SNA. | Use SNA as a reflective practice tool for professionals to assess their networks and strengthen collaborations. Assess team arrangements to identify the absence of key players or to recognize critical gaps in communication links that are necessary to work collaboratively. | Alteration and Individuals |
| Mundt 2019 [96] | USA | 143 Healthcare professionals at 5 primary clinics | Primary | Survey | Communication | Core/periphery | Clinic employees in the core of the communication network have significantly greater job satisfaction than those who are on the periphery. | To increase clinicians' job satisfaction, foster face-to-face communication among all team members. | Alteration |
| Assegaai 2019 [110] | South Africa | Community health workers (CHW) (n = 37), ward-based outreach team (WBOT) leaders (N = 3), primary healthcare facility (PHC) managers (N = 5) and local area managers (N = 2) | Community | Questionnaire | Interaction (supportive supervision) | Network graphs (indegree; density) | The supportive supervision system revolves around team leaders, who are nurse cadres and who ensure internal cohesion and support among WBOT members. The network patterns also show the extent of peer support between CHWs and WBOTs. | Relationships within teams work better than those between teams. Use SNA to identify relationships that could be strengthened. | Alteration |
| Tighe 2014 [112] | USA | A single day operating room (OR) schedule encompassing 32 anesthetizing sites | Secondary | Simulation and interviews | Interaction | Degree; betweenness; eigenvector | The OR is a scale-free network with small-world characteristics. There are differences in degree centrality between nurses and anesthesiologists and surgeons. Attendings have greater degree centrality than residents. | Use SNA to improve communication within ORs (e.g, by protecting a few highly-connected individuals; by placing senior staff into roles based on communication volumes). | Segmentation |

(Continued)

Table 3. (Continued)

| Ref | Country | Participants | Setting | Data collection | Type of tie | Network measure(s) | Key network findings | Recommendations | Network strategy |
|---|---|---|---|---|---|---|---|---|---|
| Sykes 2015 [109] | Australia | 171 Operating room (OR) staff members in 4 surgical teams in 1 hospital | Secondary | Electronic database | Interaction | Network graphs | Eighteen staff members are regularly shared across teams, including 12 nurses, five anesthetists, and one registrar. Weak but significant correlations is found between the number of staff, procedure start time, length of procedure, and patient acuity. | Use SNA to identify change champions who can support initiatives across multiple teams. | Individuals |
| Hossain 2012 [104] | USA | 204 Outpatient departments (OPDs) and 458 emergency departments (Eds) | Secondary | National survey | Coordination | Degree; Density; centralization | The nurse is the actor with highest degree, followed by physician and lab technician. There is a significant relationship between degree and performance of coordination. | Use SNA to understand the possible causes of inefficient coordination performance and coordination quality resulting in access blocks. | Alteration |
| Li 2020 [108] | China | 102 Hospital doctors | Secondary | Online forum | Communication and information exchange | Density; degree centralization; geodesic distance; centrality; reciprocity | Doctors are more closely connected, and information is easily spread. Doctors with higher professional titles show high levels of reciprocity. They are more likely to influence the behavior of other doctors. | Introduce clinical educational meetings to increase the frequency of doctor interaction at different levels. | Induction |
| Yousefi Nooraie 2017 [95] | Canada | 14 Multidisciplinary public health staff | Public health centers | Interviews | Information exchange | Indegree | Information seeking networks evolve towards more centralized structures. Staff who are already central at baseline gain even more centrality. | Use SNA to support and inform the design, process and evaluation of the evidence informed decision-making training interventions. | Induction |

(Continued)

Table 3. (Continued)

| Ref | Country | Participants | Setting | Data collection | Type of tie | Network measure(s) | Key network findings | Recommendations | Network strategy |
|---|---|---|---|---|---|---|---|---|---|
| Cannavacciuolo 2017 [113] | Romania | 28 Multidisciplinary rehabilitation unit staff. | Rehab | Questionnaire | Advice and knowledge exchange | Centrality; frequency of interactions; in-group/out-group interactions | Knowledge is shared in a centralized network characterized by the presence of a few hubs and some marginal actors. The team members consult with a high number of external experts but these sources tend to belong to personal networks and are not shared. Interpersonal knowledge exchange is mostly vertical than lateral. | Redesign the team network to improve the efficiency and effectiveness of knowledge sharing. The re-design interventions concern three main features of knowledge network: "knowledge centralization," "over-reliance on external experts", and "unshared knowledge tools and sources." Different strategies are discussed. | Alteration |

The four types of network interventions were mentioned as recommended strategies to be designed and implemented in order to improve the overall structure and functioning of the networks. Nine studies recommended to use 'individuals' [18, 22, 100, 102, 103, 106, 109, 114, 115], eight studies recommended 'induction' [24, 95, 97, 100, 106–108, 111], seven studies discussed possible 'alteration' strategies [96, 98, 104, 105, 110, 113, 114], and four recommended 'segmentation' [97, 99, 101, 112]. Four studies recommended more than one strategy [97, 100, 106, 114]. Thirteen papers were published between 2010 to 2015, and 11 between 2016 to 1st May 2022.

## Discussion

We updated previous reviews by including papers published since 2010 that have used SNA to investigate networks among healthcare professionals. Our search strategy included a wide range of databases and placed no restrictions on professional groups, healthcare setting, country, or study design. We found 102 papers that used SNA to examine networks of healthcare professionals. We confirmed the findings of prior systematic reviews: The majority of published studies were descriptive, with only four papers discussing the outcomes of an SNA-based intervention. We defined network intervention as a set of actions aimed at modifying the main elements of a network system (i.e., nodes and relations) so as to generate behavior change and improve system performance. The main idea behind network intervention is that if networks affect outcomes of interest, change in network structure could lead to change in relevant outcomes.

A possible explanation for the limited number of studies on network interventions concerns the practical difficulties in designing and implementing network-based interventions in general, and in healthcare contexts more specifically. Valente et al. [4] discuss the main challenges associated with network interventions in the domains of public health and medicine. In what follows, we will briefly describe the main challenges that we believe arise when an intervention is designed and implemented within an organizational context, such as a hospital or other healthcare organizations. Healthcare organizations present additional challenges over and above those identified by Valente et al. [4] for the public health domain. We organize our discussion by using the four-stage model of program implementation suggested by Valente et al. [4].

### Exploration

The first stage involves the assessment of a community in terms of needs, vision and opportunity for change [4]. In practice, this implies identifying: (i) a well-defined network (i.e., community boundaries); (ii) the relations among community members (i.e., social capital); (iii) the specific interests of various stakeholders, and (iv) the behavior under investigation. A number of specific challenges may arise at this stage when social network research is conducted within organizations [116]. First, network identification. This may be facilitated by the natural boundaries that organizations provide for the network of interest. Problems typically arise in collecting the non-anonymous data needed for network research. The management of the organization (which is often also the commissioner of the research) may provide partial commitment or discontinued support to the research, or even restricted access to data. Access to network and other types of data may also be problematic due to the specific nature of the population under investigation. Intervention programs within healthcare organizations are likely to involve multiple professional groups (e.g., hospital administrators, medical doctors, nurses, etc.) whose interdependencies may be difficult to manage or predict thoroughly *ex ante*. The actual use of output data from hospital administrators, participants' protection of ethical

rights, as well as the existence of ethical codes for professionals are all factors that may make data collection within healthcare organizations particularly challenging [117]. A solution to this problem may be a clear identification and communication of the goals and objectives of the research. The four studies that we identified as reporting the results of a network intervention (level-1), or those recommending a follow-up intervention in their conclusion section (level-2), mainly focused on improving specific structural features of the networks. Of the four level-1 studies, only two measured the impact of network intervention on health-related outcomes [25, 27]. The reason for this may lie in the difficulty of envisioning clear-cut causal links between behaviors at one level (e.g., health professionals) and outcomes at another level (e.g., patients). More direct evidence of measurable outcomes of network interventions at the patient or organizational level is needed. Finally, ethical challenges should also be considered at this stage. Cronin et al. [118] and Borgatti and Molina [119] offer explicit guidance on how to deal with specific ethical issues such as protecting anonymity, presenting output data in aggregated form, and offering participants multiple opportunities for opting-out.

## Adoption

The second stage involves the creation and adoption of an intervention program to address a behavioral problem [4]. The use of network analysis is particularly helpful at this stage, as it provides valuable information that can be used to tailor an intervention to the specific needs of the population under investigation. High response rates and lack of missing data are crucial as they allow to design an intervention based on more complete information. The identification of opinion leaders within a network who may act as change agents has been used in a large number of studies. Also, network analysis may be useful at this stage to identify other roles or positions, cohesive subgroups, or important cleavages within a network structure. Within an organizational setting, the existence of a formal reporting structure is particularly relevant in that it provides additional information on power structures and formal roles that can also be leveraged in a network-based intervention.

## Implementation

The third stage involves implementing the program with adherence and competence [4]. Within healthcare organizations, pressures to improve outcomes (e.g., clinical, operational, financial and managerial) are frequently generated by policy changes that produce top-down initiatives proposed by senior management and implemented through the involvement of various organizational change agents such as medical doctors, hospital administrators and, occasionally, technical and support staff. Research has recognized that the success of change initiatives hinges on the ability of change agents to overcome potential resistance from other organizational members, and encourage them to adopt or develop new practices [120]. In professional organizations, such as healthcare organizations, the coexistence of many professional groups with strong identity and role boundaries may represent the biggest obstacle to organizational change. Furthermore, not all change initiatives are equivalent, and recent research has pointed to the need of establishing the extent to which a change initiative diverges from the institutional status quo in order to better identify factors enabling adoption [120]. Other than resistance to, and extent of, change, challenges that may arise at this stage include availability of resources needed to implement a change program, lack of evidence of successful research designs to use in non-experimental, organizational settings, and lack of clarity about outcome variables to be monitored during the implementation stage.

## Sustainment

The fourth, and last stage involves checking that the program continues to be implemented as intended over time, and is continuing to exert the anticipated effects [4]. The main challenge at this stage concerns the slow-moving nature of network and organizational variables, compounded by the often-far too high turnover rates within organizational units. This could make particularly difficult predicting with a reasonable level of certainty how long a social structure would take to affect a behavior, or an outcome of interest. As this usually takes time, problems may arise that are related to changes in the composition of a network structure, which should ideally remain unchanged for the duration of an intervention program. In non-experimental, naturalistic settings this is unlikely to occur. Research has also shown that changes in the composition of a network structure led to changes in the attitudes and behaviors of those who remain in the organization [121].

We have not offered specific solutions to the various issues highlighted above. Rather, our aim was to shed light on the main challenges of implementing a change initiative within an organizational setting. A possible solution to some of the challenges associated with implementing an intervention and measuring its effects over time is the adoption of a simulation-based analytic approach. This approach involves data collected on an existing network to simulate a number of alternative scenarios resulting from altering specific characteristics of the nodes and ties within a network. An example of application of a simulation-based approach to a longitudinal network dataset can be found in Schaefer et al. [122]. The authors use the results of Stochastic Actor-Oriented Models to simulate the coevolution of friendship ties and smoking behavior under potential intervention scenarios. Currently available statistical models for network data have the advantage of being particularly well-suited for simulation analyses. This is an approach that we believe may provide realistic and interpretable evidence of the possible outcomes of a change initiative, and may justify the long-term resource commitment that network-based interventions usually require.

While a number of studies are available that describe network structure, it is important to consider that research informing on how to make positive changes in networks is likely to be closer to having practical impact. There is an urgent need for more research into which healthcare network interventions work in different contexts and how they can be best designed and employed. Similarly pressing is a need for further work to identify experimental design options that are more effective at identifying and maximizing control over relevant variables and outcomes, and that are more efficient in terms of time and resource needed. We may conclude that this is an important opportunity for the field to coalesce on terminology, measures, and applications, after establishing priority areas for researchers in how to do so to advance work on the application of SNA to the design, dissemination, implementation and sustainability of behavior change interventions.

## Limitations

We used a comprehensive broad approach to searching but may have missed some research results such as unpublished conference proceedings, papers not available in English language, negative findings or studies that did not complete and were not submitted, and grey literature.

## Conclusion

Studies of network intervention remain scant and devoid of implications for the impact of intervention initiatives on patient care. There is a need for evidence on which kinds of network interventions work, in which contexts, and under what conditions—or for whom. It is possible to measure the effect of an intervention on network effectiveness, for example, by measuring

the number of new links or increased volume of communication. However implicitly, this approach assumes a causal link between inter-professional communication and patient benefits. The complexity of healthcare, and the ubiquitous nature of barriers to best practice, implies that this is often a conjecture too far, and a more direct evidence of patient benefit should be preferred. The most important test of the effectiveness of network intervention would be assessing its impact on patient level outcomes, or, when this is difficult to determine, on the delivery of processes of care that are supported by good evidence.

## Supporting information

**S1 Table. Database results.**
(PDF)

**S2 Table. Types of network ties.**
(PDF)

**S3 Table. Network measures.**
(PDF)

**S1 File. Search strategy.**
(PDF)

**S2 File. PRISMA-ScR checklist.**
(PDF)

## Acknowledgments

**Disclaimer.** The views expressed in this publication are those of the author(s) and not necessarily those of the NIHR or the Department of Health and Social Care.

## Author Contributions

**Conceptualization:** Ameneh Ghazal Saatchi, Francesca Pallotti, Paul Sullivan.

**Data curation:** Ameneh Ghazal Saatchi.

**Formal analysis:** Ameneh Ghazal Saatchi, Francesca Pallotti, Paul Sullivan.

**Funding acquisition:** Ameneh Ghazal Saatchi.

**Investigation:** Ameneh Ghazal Saatchi, Francesca Pallotti, Paul Sullivan.

**Methodology:** Ameneh Ghazal Saatchi, Francesca Pallotti, Paul Sullivan.

**Project administration:** Ameneh Ghazal Saatchi.

**Validation:** Ameneh Ghazal Saatchi, Francesca Pallotti, Paul Sullivan.

**Writing – original draft:** Ameneh Ghazal Saatchi, Francesca Pallotti, Paul Sullivan.

**Writing – review & editing:** Ameneh Ghazal Saatchi, Francesca Pallotti, Paul Sullivan.

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
