## [Decision Letter · Decision Letter 0]

8 Nov 2022

PONE-D-22-21674Network approaches and interventions in healthcare settings: A systematic scoping reviewPLOS ONE

Dear Dr. Saatchi,

Thank you for submitting your manuscript to PLOS ONE. After careful consideration, we feel that it has merit but does not fully meet PLOS ONE’s publication criteria as it currently stands. Therefore, we invite you to submit a revised version of the manuscript that addresses the points raised during the review process.

We look forward to receiving your revised manuscript.

Kind regards,

Shahadat Uddin, PhD

Academic Editor

PLOS ONE

Journal Requirements:

Reviewers' comments:

Reviewer's Responses to Questions

**Comments to the Author**

1. Is the manuscript technically sound, and do the data support the conclusions?

Reviewer #1: Yes

Reviewer #2: Yes

2. Has the statistical analysis been performed appropriately and rigorously? 

Reviewer #1: Yes

Reviewer #2: N/A

3. Have the authors made all data underlying the findings in their manuscript fully available?

Reviewer #1: Yes

Reviewer #2: Yes

4. Is the manuscript presented in an intelligible fashion and written in standard English?

Reviewer #1: Yes

Reviewer #2: Yes

5. Review Comments to the Author

Reviewer #1: This is a systematic scoping review paper with good quality which can help researchers, healthcare providers & stakeholders and policy makers to improve the quality of healthcare delivery systems. This is also a comprehensive review paper which used network approach (SNA) to analyze the interactions between healthcare stakeholders. The literature search strategy is clearly described. The review is based on many original studies. The topic is of interest and the paper is well-written. BTW, the authors may think to explore more on the others rather than restricting SNA on networks among healthcare providers and professionals.

Reviewer #2: I would like to thank all authors for an interesting study on available evidence on SNA-based intervention programs in

healthcare settings. This review is generally well-written, structured and easy for the reader to understand. I would like to suggest to the authors to add a bit of information on the method [Preferred Reporting Items for Systematic reviews and Meta-Analyses extension for Scoping Reviews (PRISMA-ScR) Checklist] in your abstract beside addressing the Chambers’ et al. (2012) protocol.

6. PLOS authors have the option to publish the peer review history of their article (what does this mean?). If published, this will include your full peer review and any attached files.

Reviewer #1: **Yes: **SHAKIR KARIM

Reviewer #2: **Yes: **RMAG

---

## [Author Response · Author response to Decision Letter 0]

21 Jan 2023

Thank you for giving me the opportunity to submit a revised version of my manuscript titled ‘Network approaches and interventions in healthcare settings: A systematic scoping review’ to Plos One. 

We appreciate the time and effort that you and the reviewers have dedicated to providing your valuable feedback on our manuscript. We are grateful to the reviewers for their insightful comments on our paper, that have helped us improve our manuscript. We have resubmitted a revised version that address all the points raised and have found your comments insightful and valuable. We have highlighted the changes within the manuscript. Below I have responded to the reviewers’ comments point-by-point.

Comments from Reviewer #1: SHAKIR KARIM

• Comment 1: Yes

Response: Thank you for this point, we agree and therefore have taken no action.

• Comment 2: Yes

Response: Thank you for this point, we agree and therefore have taken no action.

• Comment 3: Yes

Response: Thank you for this point, we agree and therefore have taken no action.

• Comment 4: Yes

Response: Thank you for this point, we agree and therefore have taken no action.

• Comment 5: This is a systematic scoping review paper with good quality which can help researchers, healthcare providers & stakeholders and policy makers to improve the quality of healthcare delivery systems. This is also a comprehensive review paper which used network approach (SNA) to analyze the interactions between healthcare stakeholders. The literature search strategy is clearly described. The review is based on many original studies. The topic is of interest and the paper is well-written. BTW, the authors may think to explore more on the others rather than restricting SNA on networks among healthcare providers and professionals.

Response: Thank you for your valuable feedback, we are pleased you found our work to be of good quality and comprehensive. Thank you for this suggestion to look at other network relationships outside of healthcare professionals. It would have been interesting to explore this aspect. However, in the case of our study, it seems slightly out of scope because of the following reasons:

I. Chambers et al systematic scoping review focused on healthcare professionals and we wanted to provide a much-needed update as their literature search went up to 2010. Hence, to continue along this research we followed their criteria.

II. Chambers et al systematic review found only one level-1 evidence, where SNA was used as an intervention in a healthcare setting. This showed SNA on networks among healthcare professionals is complex and more evidence is needed on why we don’t have more level-1 evidence. 

III. We were able to provide a different perspective to Chambers et al by demonstrating a third level in the categorisation of networks, from those that described the network, to those that recommended a network intervention to the final level that implemented an intervention. We were therefore able to identify more level-1 evidence than Chambers et al and found a large number of evidence that suggested an intervention, which is useful for policy makers.

IV. Network interventions within organisations require individuals to implement them. Furthermore, the four network interventions mentioned by Valente et al all involve individuals and ties between individuals. 

Comments from Reviewer #2: RMAG

• Comment 1: Yes

Response: Thank you for this point, we agree and therefore have taken no action.

• Comment 2: N/A

Response: Thank you for this point, we agree and therefore have taken no action.

• Comment 3: Yes

Response: Thank you for this point, we agree and therefore have taken no action.

• Comment 4: Yes

Response: Thank you for this point, we agree and therefore have taken no action.

• Comment 5: I would like to thank all authors for an interesting study on available evidence on SNA-based intervention programs in

healthcare settings. This review is generally well-written, structured and easy for the reader to understand. I would like to suggest to the authors to add a bit of information on the method [Preferred Reporting Items for Systematic reviews and Meta-Analyses extension for Scoping Reviews (PRISMA-ScR) Checklist] in your abstract beside addressing the Chambers’ et al. (2012) protocol.

Response: Thank you for valuable feedback, we are pleased that you found the study interesting and well-written. We agree with your point on including information on the PRISMA-ScR method in the abstract to extend on our method following Chambers’ et al. (2012). Hence, we have taken your advice and added further information on our method in the abstract, in the main text and added it as a supplementory information, we hope it will be useful to you and other readers.

Additional clarifications 

In addition to the above comments, all formatting and the reference list has been checked as pointed out by the reviewers and if need be, corrected. The references needed minor correction, mainly to remove the doi’s.

We have added the tables in the main text of the manuscript and edited them to ensure they comply with the formatting requirements. Furthermore, we have renamed the supporting files and removed figure 1 from the manuscript and added it as an attachment. 

We have updated the PRISM-SCR checklist to reflect the new manuscript.

We have worked on the text within the tables to improve it and this has led to a change in the main text and supplementary files.

We look forward to hearing from you in due time regarding our submission and to respond to any further questions and comments you may have.

---

## [Decision Letter · Decision Letter 1]

7 Feb 2023

Network approaches and interventions in healthcare settings: A systematic scoping review

PONE-D-22-21674R1

Dear Dr. Saatchi,

We’re pleased to inform you that your manuscript has been judged scientifically suitable for publication and will be formally accepted for publication once it meets all outstanding technical requirements.

Kind regards,

Shahadat Uddin, PhD

Academic Editor

PLOS ONE

Additional Editor Comments (optional):

Reviewers' comments:

Reviewer's Responses to Questions

**Comments to the Author**

1. If the authors have adequately addressed your comments raised in a previous round of review and you feel that this manuscript is now acceptable for publication, you may indicate that here to bypass the “Comments to the Author” section, enter your conflict of interest statement in the “Confidential to Editor” section, and submit your "Accept" recommendation.

Reviewer #2: All comments have been addressed

2. Is the manuscript technically sound, and do the data support the conclusions?

Reviewer #2: Yes

3. Has the statistical analysis been performed appropriately and rigorously? 

Reviewer #2: N/A

4. Have the authors made all data underlying the findings in their manuscript fully available?

Reviewer #2: Yes

5. Is the manuscript presented in an intelligible fashion and written in standard English?

Reviewer #2: Yes

6. Review Comments to the Author

Reviewer #2: (No Response)

7. PLOS authors have the option to publish the peer review history of their article (what does this mean?). If published, this will include your full peer review and any attached files.

Reviewer #2: **Yes: **DR RIMAH MELATI AB.GHANI

---

## [Editor Report · Acceptance letter]

10 Feb 2023

PONE-D-22-21674R1 

Network approaches and interventions in healthcare settings: A systematic scoping review 

Dear Dr. Saatchi:

I'm pleased to inform you that your manuscript has been deemed suitable for publication in PLOS ONE. Congratulations! Your manuscript is now with our production department. 

Kind regards, 

on behalf of

Dr. Shahadat Uddin 

Academic Editor

PLOS ONE